# Basal forebrain degeneration precedes and predicts the cortical spread of Alzheimer's pathology

Taylor W. Schmitz[1,2], R. Nathan Spreng[3] & The Alzheimer's Disease Neuroimaging Initiative[†]

There is considerable debate whether Alzheimer's disease (AD) originates in basal forebrain or entorhinal cortex. Here we examined whether longitudinal decreases in basal forebrain and entorhinal cortex grey matter volume were interdependent and sequential. In a large cohort of age-matched older adults ranging from cognitively normal to AD, we demonstrate that basal forebrain volume predicts longitudinal entorhinal degeneration. Models of parallel degeneration or entorhinal origin received negligible support. We then integrated volumetric measures with an amyloid biomarker sensitive to pre-symptomatic AD pathology. Comparison between cognitively matched normal adult subgroups, delineated according to the amyloid biomarker, revealed abnormal degeneration in basal forebrain, but not entorhinal cortex. Abnormal degeneration in both basal forebrain and entorhinal cortex was only observed among prodromal (mildly amnestic) individuals. We provide evidence that basal forebrain pathology precedes and predicts both entorhinal pathology and memory impairment, challenging the widely held belief that AD has a cortical origin.

[1] Medical Research Council, Cognition and Brain Sciences Unit, 15 Chaucer Road, Cambridge CB2 7EF, UK. [2] Wolfson College, University of Cambridge, Barton Road, Cambridge CB3 9BB, UK. [3] Laboratory of Brain and Cognition, Department of Human Development, Human Neuroscience Institute, Martha Van Rensselaer Hall G62C, Cornell University, Ithaca, New York 14853, USA. Correspondence and requests for materials should be addressed to T.W.S. (email: tws35@cam.ac.uk).
[†]A full list of consortium members appears at the end of the paper.

Alzheimer's disease (AD) is a neurodegenerative disorder characterized by distributed amyloid and tau pathophysiology throughout the brain. Recent breakthroughs in molecular genetics have identified a *trans*-synaptic mechanism by which these pathologies spread across anatomically and functionally linked cortical regions[1–4] within a large-scale brain network[5–7]. These findings have potential for novel biomarkers and therapeutic strategies aimed at identifying the earliest signs of pathology and preventing its spread, before the onset of clinical AD.

However, the initial stages of AD pathophysiology remain ill defined[8,9], preventing a clear picture of what regions to target as the earliest points of spread. The prevailing model suggests that amyloid and tau deposition first appear within the transentorhinal and entorhinal cortex (EC)[10–13]. This model has been called into question by histological[14–18] and *in vivo* structural imaging evidence[19,20] of early pathological change to the nucleus basalis of Meynert (NbM) in the basal forebrain. The cholinergic cells of the NbM and their cholinoreceptive targets in EC exhibit particular sensitivity to neurofibrillary degeneration in the early stages of AD[14–18], possibly even before the onset of cognitive symptomatology[14,19]. One possible explanation for these competing findings is that the early emergence of pathology in NbM and EC occurs in parallel. A second unexplored possibility points to pathological spread from one structure to the other, indicating that NbM may constitute an earlier target of AD.

Both the NbM and EC are components of the basolateral strip, an uninterrupted band of core limbic cell groups that also includes the hippocampus, amygdala and pyriform cortex[14]. The EC receives projections from NbM and adjacent diagonal band of Broca in the primate[21,22] and human[23] brain. This primarily cholinergic innervation forms a functional pathway[24,25] involved in the encoding of novel information[26], possibly by enhancing perceptual discrimination of sensory input[27,28]. Recently, a paradoxical phenomenon of increased memory recall for task-incidental information in older adults has been linked to altered attentional modulation of sensory input at early stages of encoding[29–32], possibly arising from the loss of central cholinergic integrity[31,32]. Clarification of whether pathophysaiology in these regions manifests at the same time, or in a predictive sequence, is therefore crucial to our understanding of the early anatomical staging of AD[11] and of how this pathway may influence cognitive decline.

Given the early degeneration of NbM neurons in AD, as well as the anatomical and functional organization of the NbM—EC pathway, we predicted that NbM structural integrity would selectively determine downstream atrophy in EC. Neuroimaging biomarkers such as subregional anatomical changes in grey matter (GM) volume[33] are very highly correlated with the pathophysiological lesions of AD[34–36]. To our knowledge, measures of GM volume have not been used to track whether longitudinal changes in different regions are interdependent. In the present study, we evaluated the hypothesis of predictive pathological spread first by examining whether degeneration in NbM and EC over time exhibits interdependence and directionality. However, by itself, such a relationship does not indicate that an underlying pathology drives the interregional degenerative cascade. We therefore integrated our volumetric measures, in the same individuals, with a molecular biomarker of neuronal amyloid deposition that is extremely sensitive to AD pathophysiology at early presymptomatic stages of disease[37,38]. This strategy enabled us to test the second and critical hypothesis regarding the predictive sequence of NbM and EC degeneration across individuals at different stages of disease. Specifically, if pathology arises in NbB before spreading to EC, then degeneration in NbM and EC should dissociate at early pre-symptomatic stages of AD.

To interrogate these hypotheses, we performed longitudinal voxel-based morphometry (VBM) analyses on three high-resolution anatomical magnetic resonance imaging (MRI) brain volumes acquired over a 2-year period: T1 (baseline), T2 (1-year interval) and T3 (2-year interval) from a large, age-matched older adult sample ($N = 434$) from the Alzheimer's Disease Neuroimaging Initiative (ADNI)[39]. In addition to healthy controls (HCs; $n = 150$), the sample consisted of three groups characterized by different stages of AD as follows: mild cognitive impairment (MCI) individuals who did not progress to AD status from T1 to T3 (MCI-NP; $n = 103$), MCI individuals who progressed to AD status at T3 (MCI-AD; $n = 84$) and individuals classified as AD throughout (AD; $n = 97$). A priori regions of interest (ROIs) were specified from probabilistic anatomical maps of the EC, the Ch4 region of the basal forebrain (the magnocellular group corresponding to NbM) and a control region in the primary somatosensory cortex (PSC; see Fig. 1).

We demonstrate that baseline Ch4 volumes predicted longitudinal decreases in EC volume. By contrast, baseline EC volumes did not predict longitudinal decreases in Ch4 volume, ruling out the alternative explanations that EC precedes Ch4 degeneration, or that EC and Ch4 degeneration occurred in parallel (are mutually predictive). The predictive relationship of Ch4 volume was specific to EC: no such relationship was detected between Ch4 and the control PSC region. We next confirmed that the observed Ch4→EC predictive relationship was driven by a sequential staging of AD pathology. To do so, we used concentrations of cerebrospinal amyloid ($A\beta_{1-42}$) to distinguish, within the cognitively healthy older adult sample, individuals expressing $A\beta_{1-42}$ levels diagnostic of presymptomatic AD. We isolated an abnormal pattern of degeneration in Ch4, but not in EC, among cognitively healthy adults expressing $A\beta_{1-42}$ levels of pre-symptomatic AD. Abnormal degeneration of both Ch4 and EC was only detected at later stages of disease—among (MCI-NP) individuals expressing AD-diagnostic levels of $A\beta_{1-42}$—when short-term memory impairment was clinically detectable. Regression-based mediation and conditional process models revealed that EC degeneration mediates the relationship between Ch4 integrity and memory impairment, and that AD pathology ($A\beta_{1-42}$) moderates this mediation effect. Our results show that in the staging of AD, Ch4 degeneration precedes and predicts EC degeneration, with memory impairment emerging only after pathology spreads from Ch4 to EC. Moreover, our results suggest that abnormal Ch4 degeneration is either clinically invisible, or that the neuropsychological tests currently in widest use are not sensitive to this early subcortical stage of AD.

## Results

**GM volume changes as a function of diagnostic group.** We first examined whether and how GM volume (T1–T3) changed over the study period as a function of diagnostic group (HC, MCI-NP, MCI-AD and AD) and ROI (Ch4, EC and PSC). See Supplementary Table 1 for a complete list of ADNI individuals included, Fig. 1 for ROIs and Methods for GM volumetry). The Group and ROI factors were entered into a $4 \times 3$ repeated-measures analysis of variance (ANOVA), which included age, sex, education, head size and longitudinal changes (T1–T3) in whole-brain GM volume as covariates. We detected a significant main effect of Group ($F_{3,425} = 32.2148$, $P < 0.001$) and Group × ROI interaction ($F_{6,850} = 13.53$, $P < 0.001$; see Fig. 2a). We therefore decomposed the full factorial model into separate ANOVA models, one for each ROI, to assess the effect of clinical diagnosis on GM

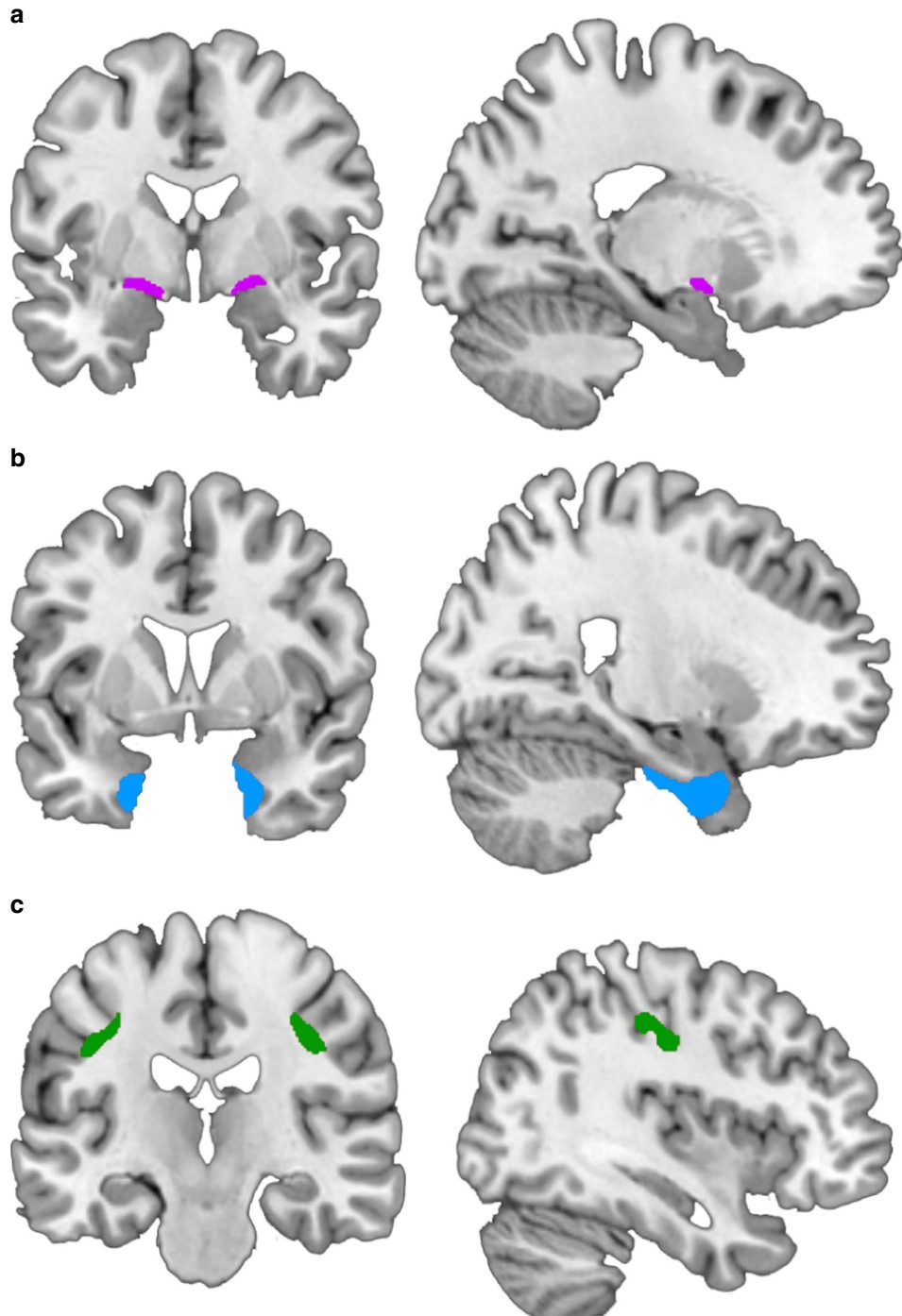

**Figure 1 | A priori ROIs.** (**a**) Basal forebrain NbM; area Ch4 (**b**) EC and (**c**) PSC. The ROIs are displayed in Montreal Neurological Institute (MNI) space on coronal (left column) and sagittal (right column) slices.

degeneration in each brain region. Magnitudes of GM degeneration in both Ch4 and EC were significantly impacted by diagnosis (Ch4: $F_{3,425} = 5.33$, $P = 0.001$; EC: $F_{3,425} = 53.39$, $P < 0.001$), which increased with clinical progression of AD. By contrast, magnitudes of GM degeneration in the PSC control region did not significantly differ as a function of diagnosis ($F_{3,425} = 2.10$, $P = 0.10$). Controlling for the same covariates, we observed a highly similar pattern of results when comparing baseline GM volume (T1) as a function of diagnostic group (HC, MCI-NP, MCI-AD and AD) and ROI (Ch4, EC and PSC). Specifically, the $4 \times 3$ ANOVA revealed a significant main effect of Group ($F_{3,425} = 49.32$, $P < 0.001$) and Group × ROI

interaction ($F_{6,850} = 18.02$, $P < 0.001$), with clinical progression to AD affecting baseline Ch4 and EC volumes more strongly than PSC (see Fig. 2b). Taken together, these initial findings confirm that our MRI measures are sensitive to local subregional, as opposed to global, anatomical changes in GM volume[33] and, moreover, that these changes increase with the clinical progression to AD.

**Parallel versus predictive spread of degeneration in Ch4 and EC.** We next assessed the competing hypotheses of parallel versus predictive spread of degeneration. To begin with, we made no

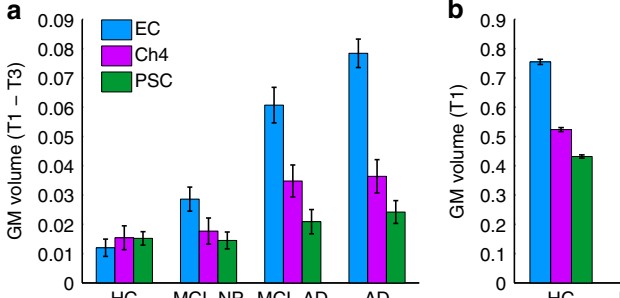
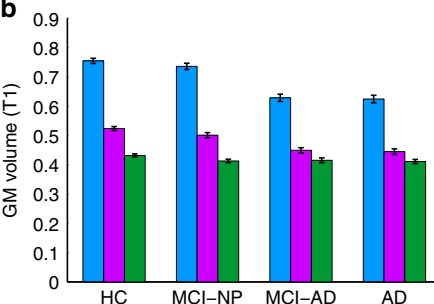

**Figure 2 | Volumetric group differences.** Volumetric group differences (diagnosis) in *a priori* ROIs for the EC (blue), basal forebrain NbM (Ch4; magenta) and PSC (green). (**a**) Magnitudes of GM degeneration from baseline (T1) to 2 years post baseline (T3) in each diagnostic group: HCs ($n = 150$), MCI–NP ($n = 103$), MCI-AD ($n = 84$) and probable AD individuals (AD; $n = 97$). (**b**) GM volume at baseline in each diagnostic group. Error bars are s.e.m.

assumptions about AD pathology in terms of diagnosis (group status) and thus used multiple regression analyses inclusive of the entire sample ($N = 434$). At baseline (T1), degeneration will already vary significantly across individuals. Smaller T1 GM volumes in EC and Ch4 should therefore reflect more advanced stages of GM degeneration before the baseline scan, after accounting for individual differences in age, sex, education, head size and whole-brain GM volume. Moreover, between time points T1 and T3 of the study period, pre-baseline degeneration is expected to progress in both EC and Ch4, but at a rate dependent on a given individual's baseline status[40]. Hence, T1 GM volume should predict the magnitude of post-baseline degeneration between time points at both 1-year (T1–T2) and 2-year (T1–T3) intervals.

According to a parallel staging model, pre-baseline degeneration in either region should predict post-baseline degeneration in the other, because both regions are affected by the same pathology at the same time. In Fig. 3a, parallel spread would be represented by superimposed degenerative trajectories in Ch4 and EC (not pictured). By contrast, according to the predictive staging model, pre-baseline degeneration in one region should only predict post-baseline degeneration in the other due to the spread of pathology over time. We contrasted the hypothesized Ch4→EC model against a competing EC→Ch4 model, where pathology originates in EC and spreads to subcortical areas of the interconnected basolateral strip (see Fig. 3a,b). In either of these scenarios, smaller T1 volumes (higher pre-baseline degeneration) in the source region should predict larger magnitudes of volumetric decrease (higher post-baseline degeneration) in the target region, yielding a negative relationship (see Fig. 3c). Based on existing models of AD progression[41–43], we predicted that the rate of change in GM degeneration would follow a nonlinear sigmoid shape. In prior volumetric MRI work, rates of GM degeneration have been shown to accelerate as patients approach clinical dementia[44,45]. A sigmoid shape as a function of time thus indicates that GM degeneration is not constant, but rather varies over the course of disease progression.

We evaluated these competing inter-regional hypotheses over both 1- and 2-year intervals. The EC→Ch4 regression model revealed virtually no relationship between pre-baseline degeneration in EC and post-baseline degeneration in Ch4 at 1-year ($t_{429} < 1$, $r = -0.03$, $P = 0.50$) and 2-year intervals ($t_{429} < 1$, $r = -0.02$, $P = 0.69$), after accounting for age, sex, education, head size and longitudinal changes in whole-brain GM volume (see Fig. 3d). By contrast, the Ch4→EC regression model yielded a significant negative relationship between pre-baseline degeneration in Ch4 and post-baseline degeneration in EC

at 1-year ($t_{429} = -4.78$, $r = -0.21$, $P < 0.001$) and 2-year intervals ($t_{429} = -5.26$, $r = -0.25$, $P < 0.001$; see Fig. 3e). Taken separately, the coefficients produced by the EC→Ch4 and Ch4→EC staging models strongly favour the Ch4→EC model, but they do not provide a direct quantitative comparison between the two models. We therefore computed a test of the equality of these two coefficients (see Methods). We found that the negative relationship observed in the Ch4→EC model was significantly stronger than the EC→Ch4 model at both 1-year ($z = 2.77$, $P = 0.003$) and 2-year intervals ($z = 3.77$, $P < 0.001$; see Fig. 3f).

We next explored the possibility that Ch4 predicts a general pattern of degeneration in neocortex, irrespective of focal susceptibility to AD or anatomical connectivity. If this were the case, pre-baseline degeneration in Ch4 should predict post-baseline degeneration even in cortical sites relatively spared by AD pathology, such as somatosensory cortex (PSC ROI). As with the direct model comparisons between Ch4 and EC (Fig. 3f), we first directly compared the Ch4→PSC model against the reverse PSC→Ch4 model, to determine whether pre-baseline degeneration in one region preferentially predicts post-baseline degeneration in the other. However, no differences were detected between these models at either 1-year ($z = 0.076$, $P_{\text{two-tailed}} = 0.94$) or 2-year intervals ($z = -1.27$, $P_{\text{two-tailed}} = 0.20$). We next directly compared the Ch4→PSC model against the observed Ch4→EC model. To do so, we computed a test of the equality of the two coefficients produced by the Ch4→EC and Ch4→PSC models (see Methods). The negative relationship produced by the Ch4→EC model was significantly stronger than the Ch4→PSC model at both 1-year ($z = -2.22$, $P_{\text{two-tailed}} = 0.03$) and 2-year intervals ($z = -2.37$, $P_{\text{two-tailed}} = 0.02$). These results suggest that Ch4 does not predict a general pattern of neocortical degeneration, but rather is selective to anatomically connected cortical targets known to be affected in the early pathological staging of AD. Together, the results from our multiple regression analyses support the hypothesis that changes in GM volume between Ch4 and EC are interdependent rather than coincidental, with pre-baseline Ch4 degeneration selectively predicting the downstream degenerative trajectory in EC.

**Amyloid staging of AD confirms Ch4 precedes EC degeneration.** Although our regression models establish evidence for a predictive sequence of GM degeneration from Ch4 to EC, by themselves they do not reveal how this pattern is influenced by AD pathology. We therefore next turned to the critical question of whether the observed pattern of Ch4→EC spread constitutes a previously unknown early link in the predictive pathological

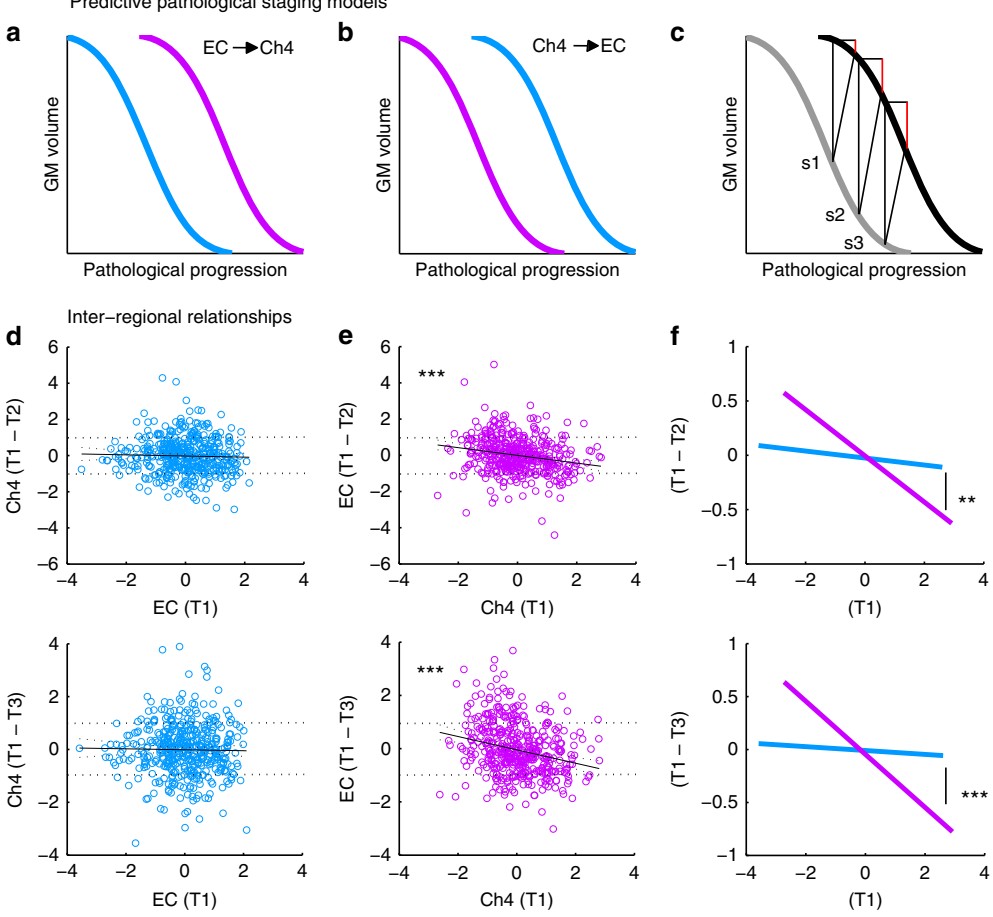

**Figure 3 | Models of degeneration and observed relationships.** (**a,b**) Hypothesized volumetric decrease (*y* axes) due to the spread of pathology over time (*x* axes) for EC (blue) and Ch4 (purple). Parallel spread model: both regions are affected by pathology at the same time (superimposed trajectories, not pictured). Predictive spread model: one region is affected by pathology before the other (offset trajectories), in either the (**a**) EC→Ch4 or (**b**) Ch4→EC model. (**c**) Pre- to post-baseline degeneration. Smaller T1 volumes in the source region of hypothetical subjects (s1—s3) predict larger volumetric decreases (T1–T3) in the target region (red lines). (**d,e**) The Ch4→EC model was significant at 1-year (middle) and 2-year intervals (bottom) across the entire sample (*n* = 434). (**f**) Direct comparisons of the predictive models at one and 2-year intervals (*y* axis range reflects the horizontal refence lines in **d,e**). Regression slopes are presented with the 95% CIs and are adjusted relative to covariates of non-interest. \*\*Two-tailed *P* < 0.01; \*\*\*two-tailed *P* < 0.001.

staging of AD. To do so, we obtained cerebrospinal fluid (CSF) measures of the Amyloid-β1 to 42 peptide ($A\beta_{1-42}$), which were available in a subset of our sample ($N = 244$). Data from the ADNI Cores were recently integrated to generate a model for the temporal ordering of AD biomarkers[38,46], which indicates that $A\beta_{1-42}$ is the first biomarker to become abnormal, followed by changes in other AD biomarkers (CSF tau, F-18 fluorodeoxyglucose-positron emission tomography) and, lastly, the onset of clinical symptoms. Crucially, receiver operating curve analysis of autopsy-confirmed AD cases versus normal controls has provided a cutpoint for CSF $A\beta_{1-42}$ concentration at which diagnostic sensitivity and specificity to AD is maximal (192 pg ml$^{-1}$), yielding correct detection of 96.4% (concentrations below 192 pg ml$^{-1}$) and correct rejection of 95.2% (concentrations above 192 pg ml$^{-1}$)[37]. We therefore first partitioned our sample into normal $A\beta$ (individuals who fell above the 192 pg ml$^{-1}$ $A\beta_{1-42}$ cutpoint) and all individuals below this cutpoint expressing AD neuropathology ($A\beta +$). Individuals presenting abnormal cognitive impairment but normal CSF $A\beta$ levels were excluded from all forthcoming analyses, as their cognitive symptoms are likely to be caused by non-AD pathology, for example, vascular dementia and hippocampal sclerosis. Of these individuals, 19 were MCI-NP, 4 were MCI-AD and 5 were AD. This left a total remaining sample size of 216:

HCs with normal $A\beta$ (HC$_{NA\beta}$: $n = 52$) and $A\beta +$ individuals ($n = 164$). We next further partitioned the $A\beta +$ group according to clinical diagnoses. This yielded the following four subgroups: individuals in the clinically silent phase of AD (HC$_{A\beta+}$: $n = 28$), MCI non-progressors (MCI-NP$_{A\beta+}$: $n = 39$), MCI-AD (MCI-AD$_{A\beta+}$: $n = 41$) and AD subgroups (AD$_{A\beta+}$: $n = 56$). This analysis strategy enabled us to explore the predictive sequence of Ch4 and EC degeneration due to AD neuropathology across individuals at difference clinical stages of AD.

The five subgroups (HC$_{NA\beta}$, HC$_{A\beta+}$, MCI-NP$_{A\beta+}$, MCI-AD$_{A\beta+}$ and AD$_{A\beta+}$) were first submitted to a $5 \times 3$ (ROI) repeated-measures ANOVA, which included age, sex, education, head size and longitudinal changes in whole-brain GM volume as covariates. This model revealed a significant Group × ROI interaction ($F_{8,412} = 6.15$, $P < 0.001$). This interaction was not dependent on GM degeneration in the PSC control region, as confirmed by a follow-up $5 \times 2$ ANOVA, which excluded this ROI ($F_{4,206} = 6.98$, $P < 0.001$). We therefore focused next on how Ch4 and EC degeneration differentially increased as a function of AD neuropathology and clinical diagnosis.

If AD neuropathology in Ch4 precedes and predicts EC, as proposed by the Ch4→EC model of predictive pathological spread, then Ch4 and EC degeneration should dissociate at early stages of disease (Fig. 4a). Consistent with this model, an

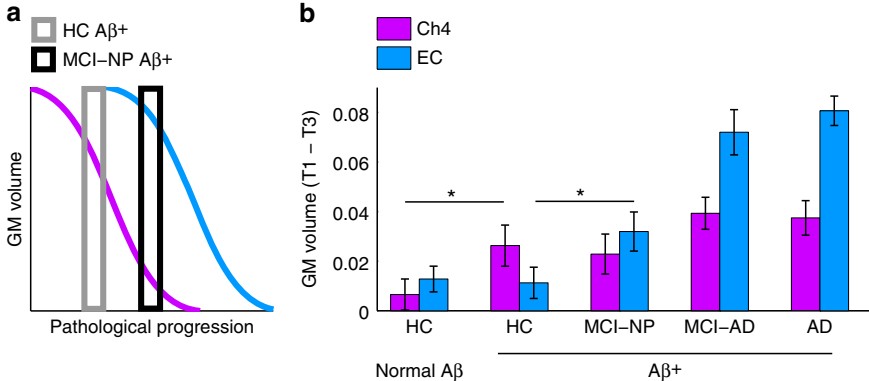

**Figure 4 | Staging GM degeneration according to the $A\beta_{1-42}$ biomarker and cognitive status. (a)** A model of predictive pathological staging in which Ch4 and EC degeneration (y axis) dissociate at early stages of pathological progression (x axis), with Ch4 degeneration emerging at stages before amnestic symptoms (HCs expressing the CSF biomarker of AD; $HC_{A\beta+}$) and degeneration in both Ch4 and EC at prodromal amnestic stages of AD (MCI non-progressors; $MCI\text{-}NP_{A\beta+}$). **(b)** Subgroups were delineated according to the CSF $A\beta_{1-42}$ cutpoint diagnostic of AD pathophysiology and then further delineated according to clinical diagnosis, yielding five subgroups: HCs with normal $A\beta$ levels ($HC_{NA\beta}$; $n = 52$), versus $HC_{A\beta+}$ ($n = 28$), $MCI\text{-}NP_{A\beta+}$ ($n = 39$), $MCI\text{-}AD_{A\beta+}$ ($n = 41$) and probable $AD_{A\beta+}$ individuals ($n = 56$). Abnormal degeneration in the $HC_{A\beta+}$ subgroup was isolated to Ch4. Abnormal degeneration in the $MCI\text{-}NP_{A\beta+}$ subgroup was observed in both Ch4 and EC. *One-tailed $P < 0.05$. Error bars are s.e.m.

### Table 1 | Biomarker staging of AD pathology in the HC group.

|  | HC subgroups | | |
|---|---|---|---|
|  | $HC_{NA\beta}$ | $HC_{A\beta+}$ | *t*-test |
| *Demographics* |  |  |  |
| Sex (male, female) | 52 (25, 27) | 28 (15, 13) | $t = 0.46$, $P = 0.65$ |
| Age (s.d.) | 75.12 (4.54) | 75.61 (5.50) | $t = 0.41$, $P = 0.69$ |
| Education (s.d.) | 15.67 (2.77) | 15.39 (3.32) | $t = 0.38$, $P = 0.71$ |
|  |  |  |  |
| *Cognitive measure* |  |  |  |
| Logical memory (immediate) | $14.72 \pm 0.37$ | $14.73 \pm 0.60$ | $t = 0.003$, $P = 0.50$ |
| Logical memory (delayed) | $13.47 \pm 044$ | $13.51 \pm 0.67$ | $t = 0.06$, $P = 0.48$ |
| RAVLT (immediate) | $8.35 \pm 0.32$ | $8.14 \pm 0.54$ | $t = 0.33$, $P = 0.37$ |
| RAVLT (delayed) | $7.62 \pm 0.40$ | $8.0 \pm 0.58$ | $t = 0.55$, $P = 0.30$ |
| RAVLT (recall) | $13.17 \pm 0.24$ | $13.05 \pm 0.34$ | $t = 0.39$, $P = 0.38$ |
| Boston Naming Test | $27.96 \pm 0.35$ | $27.81 \pm 0.42$ | $t = 0.28$, $P = 0.39$ |
| Semantic Fluency A | $20.01 \pm 0.64$ | $19.45 \pm 0.65$ | $t = 0.61$, $P = 0.27$ |
| Semantic Fluency V | $15.03 \pm 0.43$ | $14.26 \pm 0.61$ | $t = 1.03$, $P = 0.16$ |
| CDR | $0.02 \pm 0.01$ | $0.01 \pm 0.03$ | $t = 0.42$, $P = 0.34$ |

AD, Alzheimer's disease; CDR, Clinical Dementia Rating; HC, healthy control; RAVLT, Rey Auditory Verbal Learning Test.
Within subjects, values for each neuropsychological test were first averaged across the three time points of the study, to produce the most reliable estimate. Tabled values are the mean of each subgroup ± s.e.m. All *t*-statistics are independent samples *t*-tests, with 78 degrees of freedom (equal variances not assumed). Cognitive differences are assessed using a one-tailed alpha. Age and education values are in years.

independent samples *t*-test comparing the Ch4 ROI between the $HC_{NA\beta}$ and $HC_{A\beta+}$ subgroups revealed significantly larger magnitudes of GM degeneration in the clinically silent $HC_{A\beta+}$ subgroup ($t_{78} = 1.9$, $P_{1\text{-tail}} = 0.03$, $d = 0.50$). No between-group differences were detected in EC ($t_{78} = 0.19$, $P_{1\text{-tail}} = 0.85$, $d = 0.05$) or in PSC ($t_{78} = 1.07$, $P_{1\text{-tail}} = 0.16$, $d = 0.33$; see Fig. 4b). Crucially, this pattern of Ch4 degeneration was clinically silent: no differences in cognitive function between $HC_{NA\beta}$ and $HC_{A\beta+}$ subgroups were detected on any measure collected in the ADNI neuropsychological battery, even when averaging scores across study time points to produce the most stable estimates of cognitive function (see Table 1). These findings thus reveal a striking anatomical dissociation between Ch4 and EC degeneration in the clinically silent $HC_{A\beta+}$ subgroup, suggesting that EC is relatively spared alongside Ch4 atrophy within cognitively normal individuals expressing the $A\beta$ biomarker of probable AD.

As expected, we observed impairment in memory in the $MCI\text{-}NP_{A\beta+}$ compared with the $HC_{A\beta+}$ subgroup, which was most pronounced for delayed recall performance on the Logical Memory Test (see Table 2). Predictive pathological spread in the Ch4→EC pathway was thus predicted to be at a more advanced stage among $MCI\text{-}NP_{A\beta+}$ individuals, with evidence of spread to EC (Fig. 4a). Consistent with this hypothesis, independent samples *t*-test confirmed significantly larger magnitudes of EC GM degeneration in the $MCI\text{-}NP_{A\beta+}$ subgroup compared with the $HC_{A\beta+}$ subgroup ($t_{65} = 2.05$, $P_{1\text{-tail}} = 0.02$, $d = 0.51$). No between-group difference was detected in Ch4 ($t_{89} = 0.29$, $P_{1\text{-tail}} = 0.39$, $d = 0.07$), due to the abnormal elevation of Ch4 GM degeneration in both of the $HC_{A\beta+}$ and $MCI\text{-}NP_{A\beta+}$ subgroups. Moreover, no between-group difference was detected in PSC ($t_{89} = 0.04$, $P_{1\text{-tail}} = 0.48$, $d = 0.01$). In sum, compared with the isolated Ch4 pathology observed in $HC_{A\beta+}$ subgroup, we observed both Ch4 and EC pathology in the $MCI\text{-}NP_{A\beta+}$ subgroup (Fig. 4b). Taken together, these two patterns of degeneration in the Ch4→EC pathway distinguish the clinically silent and early prodromal phases of AD, strongly supporting the sequential staging of AD pathology from Ch4 to EC. By integrating volumetric and molecular biomarkers from the

ADNI, structural MRI can thus potentially be leveraged beyond its established sensitivity to early AD pathology.

At increasingly advanced stages of AD, abnormal Ch4 and EC degeneration becomes more difficult to detect against the background of global cortical degeneration. Comparing between the MCI-AD$_{A\beta+}$ and AD$_{A\beta+}$ subgroups, magnitudes of GM degeneration were indistinguishable in both Ch4 ($t_{95} = 0.19$, $P_{1\text{-tail}} = 0.85$, $d = 0.04$) and EC ($t_{95} = 0.80$, $P_{1\text{-tail}} = 0.43$, $d = 0.19$), despite being substantially elevated relative to the HC$_{NA\beta}$, HC$_{A\beta+}$ and MCI-NP$_{A\beta+}$ subgroups (Fig. 4b). This observation serves to highlight the importance of both EC and Ch4 as important early indicators of pathological change, which lose their viability as biomarkers with the current clinical tools used to detect AD progression.

**A hypothesized subcortical–cortical pathway to AD.** Thus far, we have shown evidence that GM integrity in Ch4 predicts subsequent GM degeneration in EC, and that abnormal Ch4 GM degeneration precedes both memory impairment and abnormal EC degeneration in the pathological staging of AD. These findings suggest a subcortical–cortical pathway to AD, with the spread of pathology from Ch4 to EC inducing selective impairments in memory recall for novel information. We therefore next examined with mediation analyses[47] how the regression-based evidence of Ch4→EC predictive pathological spread may relate to memory dysfunction. Critically, we then used conditional process analysis to determine whether these relationships are dependent on the presence of AD neuropathology (see Methods).

To do so, we first used multiple linear regression analysis of the entire study sample ($N = 434$) to determine whether a relationship existed between pre-baseline degeneration in Ch4 (T1 volume) and memory recall performance. The latter was indexed from the Logical Memory Test delayed recall score, given its observed sensitivity to early impairments in the MCI-NP$_{A\beta+}$ group (see Table 2). We found a significant positive relationship ($r = 0.33$, $t = 7.30$, $P < 0.001$), whereby smaller Ch4 GM volumes predicted lower recall performance, after accounting for age, sex, education, head size and longitudinal changes in whole-brain GM volume. However, our findings from the A$\beta_{1-42}$ staging of Ch4 and EC degeneration indicate that deficits in memory recall

manifest only once pathological spread from Ch4 to EC has occurred. If this is the case, then the observed relationship between Ch4 atrophy and impaired recall should be better accounted for by modelling the Ch4→EC pathway. We tested this hypothesis using a mediation analysis, which included post-baseline EC degeneration (T1–T3) as an indirect pathway between Ch4 and delayed recall. Path $a$ in the mediation model thus replicates the prior Ch4→EC regression model. The same nuisance covariates used in the linear regression were included in the mediation model (see Methods). The bootstrapped unstandardized indirect effect was 6.76, with the 95% confidence interval (95% CI) ranging from 4.22 to 9.68, indicating a significant mediation. We further confirmed the significance of this mediation effect using the Sobel test ($z = 4.73$, $P < 0.001$). We also confirmed the anatomical specificity of this mediation effect by performing a second control analysis, substituting EC with PSC. This single alteration to the model abolished the mediation effect (unstandardized indirect effect $= 0.26$, 95% CI ($-0.19$ to $1.33$); Sobel $z = 0.75$, $P = 0.46$).

Our Ch4→EC regression and mediation results demonstrate that pathology originating in Ch4 gives rise to memory dysfunction through predictive spread to EC. To confirm that the mediation effect is indeed dependent on AD neuropathology, we returned to the subsample of individuals in whom CSF A$\beta$ measures were available ($N = 216$). We first confirmed that the regression and mediation findings observed in the whole sample held in the subsample (regression model: $r = 0.32$, $t = 4.84$, $P < 0.001$; mediation model: unstandardized indirect effect $= 8.60$, 95% CI (4.93 to 12.99), Sobel $z = 3.86$, $P < 0.001$; PSC control mediation model: unstandardized indirect effect $= 0.67$, 95% CI ($-0.21$ to $3.01$); Sobel $z = 0.93$, $P = 0.35$; see Fig. 5a,b). We next re-partitioned the subsample according to those who fell below or above the 192 pg ml$^{-1}$ A$\beta_{1-42}$ cutpoint at which AD diagnostic accuracy is maximal[37], that is, the HC$_{NA\beta}$ ($N = 52$) and A$\beta+$ ($N = 164$) groups. The two groups were coded as a dichotomous moderator variable. We were thus able to determine whether the observed Ch4→EC→Recall mediation effect was moderated by the presence of AD neuropathology. Specifically, we hypothesized that the A$\beta+$ group would drive the mediation effect by increasing the strength of the relationships on both the Ch4→EC path ($a$) and the EC→Recall path ($b$). As before, we included age, sex, education, head size and longitudinal changes in whole-brain

**Table 2 | Biomarker staging of AD pathology in the HC and MCI-NP groups.**

| | Subgroups | | |
| --- | --- | --- | --- |
| | HC$_{A\beta+}$ | MCI-NP$_{A\beta+}$ | *t*-test |
| *Demographics* | | | |
| Sex (male, female) | 28 (15, 13) | 39 (24, 15) | $t = 1.29$, $P = 0.20$ |
| Age (s.d.) | 75.61 (5.50) | 73.05 (6.96) | $t = 1.67$, $P = 0.10$ |
| Education (s.d.) | 15.39 (3.32) | 16.26 (2.71) | $t = 1.13$, $P = 0.26$ |
| | | | |
| *Cognitive measure* | | | |
| Logical memory (immediate) | 14.73 ± 0.60 | 7.51 ± 0.43 | $t = 9.72$, $P < 0.001$ |
| Logical memory (delayed) | 13.51 ± 0.67 | 4.27 ± 0.52 | $t = 10.86$, $P < 0.001$ |
| RAVLT (immediate) | 8.14 ± 0.54 | 3.79 ± 0.40 | $t = 6.44$, $P < 0.001$ |
| RAVLT (delayed) | 8.0 ± 0.58 | 2.77 ± 0.45 | $t = 7.12$, $P < 0.001$ |
| RAVLT (recall) | 13.05 ± 0.34 | 9.64 ± 0.47 | $t = 5.88$, $P < 0.001$ |
| Boston Naming Test | 27.81 ± 0.42 | 26.01 ± 0.84 | $t = 1.91$, $P = 0.03$ |
| Semantic Fluency A | 19.45 ± 0.65 | 15.63 ± 0.68 | $t = 4.05$, $P < 0.001$ |
| Semantic Fluency V | 14.26 ± 0.61 | 10.57 ± 0.51 | $t = 4.66$, $P < 0.001$ |
| CDR | 0.01 ± 0.03 | 0.49 ± 0.01 | $t = 14.87$, $P < 0.001$ |

AD, Alzheimer's disease; CDR, Clinical Dementia Rating; HC, healthy control; MCI-NP, mild cognitive (who did not progress to probable AD); RAVLT, Rey Auditory Verbal Learning Test.
Within subjects, values for each neuropsychological test were first averaged across the three time points of the study, to produce the most reliable estimate. Tabled values are the mean of each subgroup ± s.e.m. All *t*-statistics are independent samples *t*-tests, with 65 degrees of freedom (equal variances not assumed). Cognitive differences are assessed using a one-tailed alpha. Age and education values are in years.

GM volume as covariates. Consistent with our hypotheses, a significant mediation effect was detected in the $A\beta+$ subgroup (unstandardized indirect effect = 6.11, 95% CI (2.49 to 10.48)). The mediation effect was abolished altogether in the $HC_{NA\beta}$ subgroup (standardized indirect effect = − 1.90, 95% CI (− 6.78 to 0.30)). A direct test of the equality between these conditional indirect effects in each subgroup confirmed that they differed significantly (unstandardized moderation effect = − 8.01, 95% CI (− 13.81 to − 3.46); see Fig. 5c). We therefore provide evidence of a conditional process model for early predictive pathological staging in the Ch4→EC pathway. This model links the Ch4→EC degenerative sequence to memory dysfunction and reveals that this process is dependent on AD neuropathology.

## Discussion

In sum, our findings build on a varied body of histological evidence, which collectively point to both Ch4 and EC as early targets of AD pathology[8]. The cell groups constituting these structures are among the first to express intraneuronal neurofibrilliary tangles and $A\beta$-containing plaques in cognitively normal older adults. They are also among the cell groups most devastated by tangles and plaques in MCI and AD[8,11,14]. The proliferation of tangles and plaques in these structures leads to depletion of axons and cell loss, both of which contribute to microstructural decreases in volume[8,15,17,18]. Our *in vivo* structural data are sensitive to these volumetric changes, which we confirmed through regression analyses with the CSF $A\beta_{1-42}$ biomarker of amyloid pathology. Strikingly, by leveraging the biomarker data to identify cognitively normal individuals expressing AD neuropathology, we show that abnormal changes in Ch4 GM volume were apparent even in clinically silent stages of probable AD.

Several lines of evidence have diminished the role of cholinergic degradation in the cognitive dysfunctions of AD[9,48–50], which appears at odds with the histological evidence emphasizing a cholinergic lesion in AD pathology. Our findings reconcile the contention that Ch4 is at once a central pathological target of AD, but also that Ch4 pathology by itself does not account for the memory impairments observed in AD[9]. We isolated abnormal degeneration to Ch4 in the clinically silent stage ($HC_{A\beta+}$), before cognitive symptoms were detectable on any of the ADNI neuropsychological measures. Memory dysfunction manifested only later in the progression of AD, among MCI-$NP_{AB+}$ individuals expressing abnormal degeneration in both Ch4 and EC. Hence, it is a lesion of the Ch4→EC pathway, causally induced by predictive pathological spread, which gives rise to the memory dysfunctions observed in early AD. This interpretation is consistent with animal work showing that neurotoxic lesions to either Ch4 or EC yield only moderate impairments in memory, whereas lesions to both structures yields a dramatic deficit in the ability to acquire new memories and cause behavioural disturbances that mimic the restlessness and wandering observed in AD[51]. In a separate line of neurophysiological research, cholinergic Ch4 projections have been proposed to tune the oscillatory dynamics of EC neurons during memory encoding[24,25]. Confirming this hypothesis, selective lesions to the cholinergic innervations of EC were subsequently shown to impair working memory for novel but not familiar stimuli[26]. Memory encoding thus depends on the functional integrity of both Ch4 and EC. Damage to both of these structures, or their connections, yields a selective memory impairment highly consistent with the anterograde amnesia observed in prodromal stages of AD[52] (see Table 2).

One question raised by the present study is whether the clinically silent phase of AD, during which pathophysiology is restricted to Ch4, is indeed clinically silent. An emerging pattern in the cognitive ageing literature indicates that some healthy, older adults more than others exhibit unique deficits of feature-selective attention—that is, the capacity to suppress unattended features of sensory input—yielding a greater susceptibility to incidental encoding of stimuli such as visual distractors[29–32]. Population neural coding of sensory information in the visual cortex is strongly linked to the basal forebrain cholinergic system, in both human[53–56] and animal[27,28,57–59] research. The stimulus-driven pattern of encoding in older adulthood may therefore constitute an early pathological sign of AD due to loss of cortical cholinergic neurons in the basal forebrain. Precise behavioural measures of feature-selective attention, sensitive to early cholinergic dysfunction in older adults, have the potential to provide clinical measures sensitive to early stages of AD that precede memory impairment.

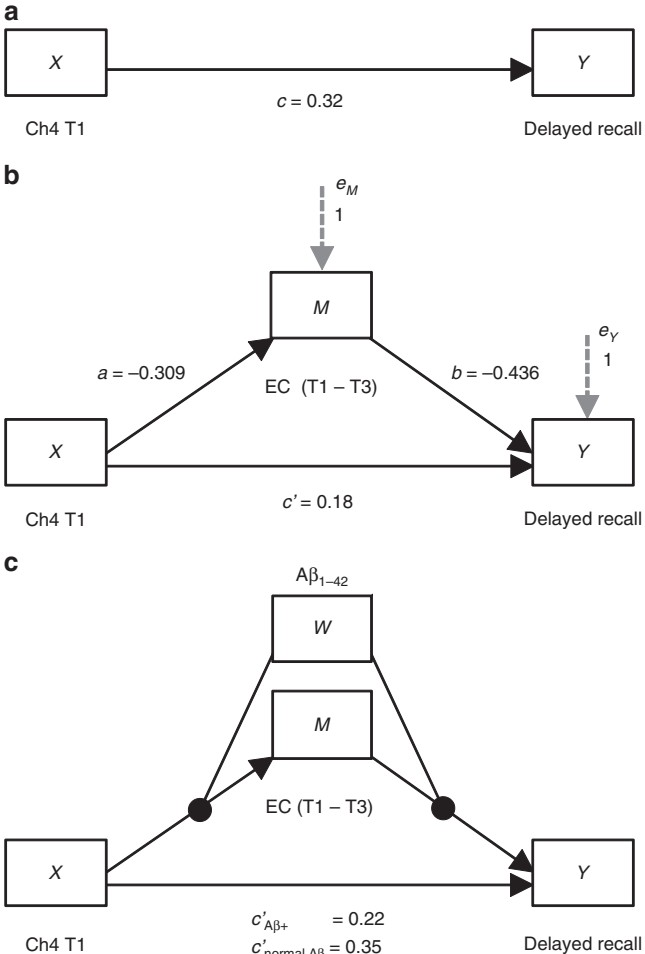

**Figure 5 | Mediation and conditional process analysis of Ch4→EC predictive pathological staging.** (**a**) Multiple linear regression analysis confirmed that a relationship exists between smaller Ch4 volume and lower delayed recall performance ($n = 216$). (**b**) Mediation analysis revealed that Ch4 volume better predicts recall performance when accounting for longitudinal EC degeneration, that is, the direct relationship (path $c'$) is suppressed. (**c**) Conditional process analysis demonstrated that the observed mediation effect was moderated by the CSF $A\beta$ biomarker of AD neuropathology: the Ch4→EC→Recall mediation effect was significant for the $A\beta+$ individuals ($n = 164$), that is, path $c'$ is suppressed, but not for individuals with normal $A\beta$ ($n = 52$). All path coefficients were significant two-tailed $P < 0.001$. Error terms for the mediation ($e_m$) and criterion variables ($e_y$) are denoted with grey hashed arrows.

**Table 3 | Demographic information from the ADNI.**

| | Diagnostic group | | | |
|---|---|---|---|---|
| | **HC** | **MCI-NP** | **MCI-AD** | **AD** |
| N (male, female) | 150 (77, 73) | 103 (69, 34) | 84 (52, 32) | 97 (49, 48) |
| Age (s.d.) | 75.6 (4.9) | 74.6 (6.9) | 74.4 (6.8) | 74.5 (7.3) |
| Education (s.d.) | 15.9 (2.9) | 16.0 (2.8) | 15.7 (3.2) | 14.8 (2.9) |
| Handed (R,L) | 138, 12 | 96, 7 | 78, 6 | 91, 6 |
| | | | | |
| MMSE | | | | |
| T1 | 29.2 (0.97) | 27.6 (1.71) | 25.9 (2.12) | 23.1 (1.92) |
| T2 | 29.2 (1.13) | 27.7 (2.53) | 23.2 (3.04) | 20.6 (4.58) |
| T3 | 29.2 (1.02) | 27.3 (3.25) | 21.9 (4.61) | 18.3 (6.14) |

AD, probable Alzheimer's disease; ADNI, Alzheimer's Disease Neuroimaging Initiative; HC, healthy control; MCI-AD, mild cognitive impairment (who progressed to probable AD); MCI-NP, mild cognitive impairment (who did not progress to probable AD); MMSE, Mini-mental State Exam; T, timepoint from the longitudinal sample.
Summary of subject demographics for each diagnostic category in the ADNI sample. Age and education values are in years.

The present study demonstrates that changes in Ch4 GM volume predict and precede both memory impairment and longitudinal changes in EC GM volume, and critically that these relationships are dependent on AD neuropathology. Among individuals expressing the $A\beta_{1-42}$ CSF biomarker of probable AD, patterns of abnormal degeneration in the Ch4→EC pathway can dissociate those with memory impairments from those who appear cognitively normal. Specifically, a pattern of isolated abnormal Ch4 degeneration was observed among clinically silent probable AD individuals ($HC_{A\beta+}$), with selective memory impairment emerging only once this pattern of abnormal degeneration affected both Ch4 and EC ($MCI-NP_{A\beta+}$). One limitation of this study is the relatively small number of $HC_{A\beta+}$ individuals available through ADNI, whose measures of CSF A$\beta$ can be integrated with longitudinal structural MRI and neuropsychological data ($n = 28$). Our findings indicate that future large-scale research initiatives on AD would benefit from a multimodal biomarker strategy including, at a minimum, CSF A$\beta$ and longitudinal structural MRI, focused on cognitively healthy adults. Nevertheless, our findings strongly suggest that a subcortical–cortical pathological spread from Ch4 to EC defines the earliest link in the predictive pathological staging of AD.

Although our imaging results complement recent breakthroughs in molecular genetics showing that AD spreads via a *trans*-synaptic mechanism[1–4], they also necessitate reconsideration of EC as the origin point of disease. Molecular genetics holds promise for developing therapeutic strategies to prevent the spread of pathology at stages of AD preceding even the earliest memory impairments. Our evidence strongly indicates that these efforts will be more effective if they target the basal forebrain rather than EC.

## Methods

**ADNI participants and MRI acquisition.** Data used in the preparation of this article were obtained from the ADNI database (adni.loni.usc.edu). The ADNI was launched in 2003 as a public–private partnership, led by Principal Investigator Michael W. Weiner, M.D. The primary goal of ADNI has been to test whether serial MRI, positron emission tomography, other biological markers, and clinical and neuropsychological assessment can be combined to measure the progression of MCI and early AD. Determination of sensitive and specific markers of very early AD progression is intended to aid researchers and clinicians, to develop new treatments and monitor their effectiveness, as well as lessen the time and cost of clinical trials. Subjects were recruited from 50 sites in the United States and Canada. Written informed consent was obtained from all participants before protocol-specific procedures were performed. All data acquired as part of this study are publicly available. For up-to-date information, see www.adni-info.org.

In the present study, we included ADNI1 participants ($N = 434$, 187 women, see Supplementary Table 1 and Table 3, downloaded 5 October 2012) with three time points, separated by at least 12 months, and no more than 30 months. Participants were divided into four groups: HCs, MCI-NP to AD and MCI patients who did progress to AD (MCI-AD) and probable AD patients. To assess brain

change covering the transition from MCI to AD, we anchored the longitudinal analysis to the scan in which MCI participants progressed to AD. Therefore, MCI-AD patients always transitioned to AD at time point 2. We then defined the study window as 12 months (minimum 6 months) before progression and 12 months (maximum 18 months) post progression. This 24-month study window was replicated for the other three cohorts from the baseline scan and the two subsequent annual follow-up scans (12 months and 24 months). Healthy participants who subsequently progressed to AD or MCI were excluded ($n = 10$). All participant neuroimage ID numbers are in Supplementary Table 1. Baseline diagnostic status was assessed with the Mini-Mental Status Examination (Table 3), Wechsler Memory Scale (Logical Memory subtest), Clinical Dementia Rating Scores, in addition to subjective reports. A probable AD diagnosis was made following NINCDS/ADRDA criteria. Information on recruitment and diagnostic criteria can be found on the ADNI website: www.adni-info.org.

MRI data were collected according to a standardized protocol (Jack et al.[60]). This protocol included a high-resolution T1-weighted, rapid gradient echo sequence on a 1.5 T scanner. The ADNI MRI Core optimized acquisition parameters of the neuroimage sequences for each scanner make and model. Sample high-resolution T1-weighted, rapid gradient echo acquisition parameters for one platform (Siemens Magnetom Sonata syngo MR 2004 A) was as follows: T1 = 1,000 ms, TR = 2,400 ms, TE = minimum, flip angle = 88, bandwidth 180 Hz per pixel, FOV = 240 mm, matrix size = 192 × 192, 60 slices and slice thickness = 1.2 mm. All data correction and neuroimage quality-control procedures were performed at the Mayo Clinic. Neuroimage quality control included inspection for protocol compliance, clinically significant medical abnormalities and neuroimage quality. To enhance standardization across ADNI sites, post-acquisition correction of neuroimage artefacts was also implemented. This included corrections in geometry for gradient nonlinearity, intensity non-uniformity due to non-uniform receiver coil sensitivity or additional causes[60]. Consistent with the formulation of standardized data sets[61], participant scans were included in the current study if the MRI of one of the two T1 anatomical scans passed the quality-control process.

**MRI data preprocessing.** All neuroimages were preprocessed in SPM8 using the diffeomorphic anatomical registration through exponentiated lie algebra (DAR-TEL)[62] and longitudinal VBM8 toolboxes (http://dbm.neuro.uni-jena.de/vbm8/). Anatomical images were segmented into the GM, white matter, cerebral spinal fluid, bone and soft tissue. GM neuroimages were realigned within subject, then normalized to a population template in Montreal Neurological Institute space. All neuroimages were then subjected to non-linear modulation that plotted the absolute amount of brain tissue, corrected for participant head size in VBM8. Neuroimages were then sampled with a resulting voxel size 1.5 mm$^3$. Total intracranial volume (that is, head size) was computed as the sum of the GM, white matter and CSF volumes derived from non-normalized segmented images.

**ROI analysis.** ROIs for NbM and EC were defined from probabilistic maps using the SPM Anatomy Toolbox[63], to ensure anatomical precision and replicability (see Fig. 1 and Methods). The basal forebrain is composed of distinct magnocellular cholinergic cell groups, defined histologically in non-human primates as Ch1–Ch6 (ref. 22), with Ch4 corresponding to NbM. A stereotaxic probabilistic anatomical map of Ch4 was recently obtained in humans from postmortem brains[64]. We refer to the NbM ROI as Ch4 henceforth, to reflect this anatomical parcellation. The EC ROI was similarly obtained from an existing stereotaxic probabilistic anatomical map[65]. We also obtained measures of GM volume from a third ROI in the PSC, which we used as a control to confirm the anatomical specificity of predictive pathological staging. Alzheimer's is characterized by a relative sparing of PSC and a lack of somatosensory symptomatology[12,66]. The PSC was derived from a stereotaxic

probabilistic anatomical map of somatosensory area 3a, which lies at the fundus of the central sulcus[67].

Probability maps were calculated from postmortem histological analyses based on a sample of ten brains. Each map describes the relative frequency at which the same area (for example, Ch4 EC or PSC) was represented in each voxel of the reference space. The ROIs were created by applying a threshold of 50% to the corresponding probability map. Thus, for the ROIs, only those voxels were considered that were present in more than five postmortem brains. The ROIs were linearly coregistered with the modulated GM images in Montreal Neurological Institute space. To produce indices of longitudinal degeneration, for each participant we subtracted their unsmoothed modulated GM images at T2 (short interval) or at T3 (long interval) from their unsmoothed modulated GM image at T1 using custom Matlab scripts. Within each ROI, values for mean GM volume and longitudinal degeneration were extracted using the Marsbar toolbox[68] and custom Matlab scripts.

**Regression analysis.** All multiple linear regression analyses employed robust estimation, thereby minimizing potential outlier effects. Statistical tests of the equality between two dependent correlations with no variables in common were performed as follows: first, each correlation coefficient was converted into a z-score using Fisher's r-to-z transformation. Then, equations (2) and (11) from Steiger[69] were used to compute the asymptotic covariance of the estimates. These quantities were used in an asymptotic z-test. Statistical tests of the equality between two dependent correlations with one variable in common were performed in the same manner, except that equations (3) and (10) from Steiger[69] were used to compute the asymptotic covariance of the estimates.

The mediation and conditional process analyses employed a regression-based path analytic framework for estimating direct and indirect effects[47,70]. The dependent measure, Logical Memory Test delayed recall, was first averaged within subjects across the three study time points to produce the most reliable estimate of performance. For the mediation model, the standarized indirect effect was determined by multiplying paths $a(-0.309)$ and $b(-0.436)$, yielding 0.135. For both the mediation and conditional process models, inference of statistical significance for the conditional indirect effects was determined using bias-corrected bootstrapping procedures. Specifically, unstandardized indirect effects were computed for each of 10,000 bootstrapped samples and the 95% CI was computed by determining the indirect effects at the 2.5th and 97.5th percentiles.

**Data availability.** All data used in this study are available from the ADNI database (adni.loni.usc.edu).

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

## Acknowledgements

We thank Elizabeth DuPre and Marieke C. Mur for advice and assistance. This work was supported in part by an Alzheimer's Association grant (NIRG-14-320049) to R.N.S. Data used in preparation of this article were obtained from the ADNI database (adni.loni.usc.edu). As such, the investigators within the ADNI contributed to the design and implementation of ADNI and/or provided data, but did not participate in analysis or writing of this report. A complete listing of ADNI investigators can be found at: http://adni.loni.usc.edu/wp-content/uploads/how_to_apply/ADNI_Acknowledgement_List.pdf. Data collection and sharing for this project were funded by the Alzheimer's Disease Data collection and sharing for this project was funded by the ADNI (National Institutes of Health Grant U01 AG024904) and DOD ADNI (Department of Defense award number W81XWH-12-2-0012). ADNI is funded by the National Institute on Aging, the National Institute of Biomedical Imaging and Bioengineering, and through generous contributions from the following: AbbVie, Alzheimer's Association; Alzheimer's Drug Discovery Foundation; Araclon Biotech; BioClinica, Inc.; Biogen; Bristol-Myers Squibb Company; CereSpir, Inc.; Eisai Inc.; Elan Pharmaceuticals, Inc.; Eli Lilly and Company; EuroImmun; F. Hoffmann-La Roche Ltd and its affiliated company Genentech, Inc.; Fujirebio; GE Healthcare; IXICO Ltd; Janssen Alzheimer Immunotherapy Research and Development, LLC; Johnson & Johnson Pharmaceutical Research and Development LLC; Lumosity; Lundbeck; Merck & Co., Inc.; Meso Scale Diagnostics, LLC; NeuroRx Research; Neurotrack Technologies; Novartis Pharmaceuticals Corporation; Pfizer, Inc.; Piramal Imaging; Servier; Takeda Pharmaceutical Company; and Transition Therapeutics. The Canadian Institutes of Health Research is providing funds to support ADNI clinical sites in Canada. Private sector contributions are facilitated by the Foundation for the National Institutes of Health (www.fnih.org). The grantee organization is the Northern California Institute for Research and Education and the study is coordinated by the Alzheimer's Disease Cooperative Study at the University of California, San Diego. ADNI data are disseminated by the Laboratory for Neuro Imaging at the University of Southern California.

## Author contributions

ADNI collected the data. R.N.S. preprocessed the data. T.W.S. conducted the analyses. T.W.S. and R.N.S. wrote the paper.

## Additional information

**Competing financial interests:** The authors declare no competing financial interests.

## The Alzheimer's Disease Neuroimaging Initiative

Michael W. Weiner[4], Paul Aisen[5], Ronald Petersen[6], Clifford R. Jack[6], William Jagust[7], John Q. Trojanowski[8], Arthur W. Toga[9], Laurel Beckett[10], Robert C. Green[11], Andrew J. Saykin[12], John Morris[13], Leslie M. Shaw[8], Zaven Khachaturian[5,14], Greg Sorensen[15], Lew Kuller[16], Marc Raichle[13], Steven Paul[17], Peter Davies[18], Howard Fillit[19], Franz Hefti[20], Davie Holtzman[13], M. Marcel Mesulam[21], William Potter[22], Peter Snyder[23], Adam Schwartz[24], Tom Montine[25], Ronald G. Thomas[5], Michael Donohue[5], Sarah Walter[5], Devon Gessert[5], Tamie Sather[5], Gus Jiminez[5], Danielle Harvey[10], Matthew Bernstein[6], Nick Fox[26], Paul Thompson[27], Norbert Schuff[4,10], Bret Borowski[6], Jeff Gunter[6], Matt Senjem[6], Prashanthi Vemuri[6], David Jones[6], Kejal Kantarci[6], Chad Ward[6], Robert A. Koeppe[28], Norm Foster[29], Eric M. Reiman[30], Kewei Chen[30], Chet Mathis[16], Susan Landau[7], Nigel J. Cairns[13], Erin Householder[13], Lisa Taylor-Reinwald[13], Virginia Lee[8], Magdalena Korecka[8], Michal Figurski[8], Karen Crawford[9], Scott Neu[9], Tatiana M. Foroud[12], Steven Potkin[31], Li Shen[12], Kelley Faber[12], Sungeun Kim[12], Kwangsik Nho[12], Leon Thal[5], Neil Buckholtz[32], Marylyn Albert[33], Richard Frank[34], John Hsiao[32], Jeffrey Kaye[35], Joseph Quinn[35], Betty Lind[35], Raina Carter[35], Sara Dolen[35], Lon S. Schneider[9], Sonia Pawluczyk[9], Mauricio Beccera[9], Liberty Teodoro[9], Bryan M. Spann[9], James Brewer[5], Helen Vanderswag[5], Adam Fleisher[5,30], Judith L. Heidebrink[28], Joanne L. Lord[28], Sara S. Mason[6], Colleen S. Albers[6], David Knopman[6], Kris Johnson[6], Rachelle S. Doody[36], Javier Villanueva-Meyer[36], Munir Chowdhury[36], Susan Rountree[36], Mimi Dang[36], Yaakov Stern[36], Lawrence S. Honig[36], Karen L. Bell[36], Beau Ances[13], Maria Carroll[13], Sue Leon[13], Mark A. Mintun[13], Stacy Schneider[13], Angela Oliver[13], Daniel Marson[37], Randall Griffith[37], David Clark[37], David Geldmacher[37], John Brockington[37], Erik Roberson[37], Hillel Grossman[38], Effie Mitsis[38], Leyla de Toledo-Morrell[39], Raj C. Shah[39], Ranjan Duara[40], Daniel Varon[40], Maria T. Greig[40], Peggy Roberts[40], Marilyn Albert[33], Chiadi Onyike[33], Daniel D'Agostino[33], Stephanie Kielb[33], James E. Galvin[41], Brittany Cerbone[41], Christina A. Michel[41], Henry Rusinek[41], Mony J. de Leon[41], Lidia Glodzik[41], Susan De Santi[41], P. Murali Doraiswamy[42], Jeffrey R. Petrella[42], Terence Z. Wong[42], Steven E. Arnold[8], Jason H. Karlawish[8], David Wolk[8], Charles D. Smith[43], Greg Jicha[43], Peter Hardy[43], Partha Sinha[43], Elizabeth Oates[43], Gary Conrad[43], Oscar L. Lopez[16], MaryAnn Oakley[16], Donna M. Simpson[33], Anton P. Porsteinsson[44], Bonnie S. Goldstein[44], Kim Martin[44], Kelly M. Makino[44], M. Saleem Ismail[44], Connie Brand[44], Ruth A. Mulnard[31], Gaby Thai[31], Catherine Mc-Adams-Ortiz[31], Kyle Womack[45], Dana Mathews[45], Mary Quiceno[45], Ramon Diaz-Arrastia[45], Richard King[45], Myron Weiner[45], Kristen Martin-Cook[45], Michael DeVous[45], Allan I. Levey[46], James J. Lah[46], Janet S. Cellar[46], Jeffrey M. Burns[47], Heather S. Anderson[47], Russell H. Swerdlow[47], Liana Apostolova[27], Kathleen Tingus[27], Ellen Woo[27], Daniel H.S. Silverman[27], Po H. Lu[27], George Bartzokis[27], Neill R. Graff-Radford[48], Francine Parfitt[48], Tracy Kendall[48], Heather Johnson[48], Martin R. Farlow[12], AnnMarie Hake[12], Brandy R. Matthews[12], Scott Herring[12], Cynthia Hunt[12], Christopher H. van Dyck[49], Richard E. Carson[49], Martha G. MacAvoy[49], Howard Chertkow[50], Howard Bergman[50], Chris Hosein[50], Sandra Black[51], Bojana Stefanovic[51], Curtis Caldwell[51], Ging-Yuek Robin Hsiung[52], Howard Feldman[52], Benita Mudge[52], Michele Assaly[52], Andrew Kertesz[53,54,55], John Rogers[53,55], Charles Bernick[53], Donna Munic[53], Diana Kerwin[21], Marek-Marsel Mesulam[21], Kristine Lipowski[21], Chuang-Kuo Wu[21], Nancy Johnson[21], Carl Sadowsky[56], Walter Martinez[56], Teresa Villena[56], Raymond Scott Turner[57], Kathleen Johnson[57], Brigid Reynolds[57], Reisa A. Sperling[11], Keith A. Johnson[11], Gad Marshall[11], Meghan Frey[11], Barton Lane[11], Allyson Rosen[11], Jared Tinklenberg[11], Marwan N. Sabbagh[58], Christine M. Belden[58], Sandra A. Jacobson[58], Sherye A. Sirrel[58], Neil Kowall[59], Ronald Killiany[59], Andrew E. Budson[59], Alexander Norbash[59], Patricia Lynn Johnson[59], Joanne Allard[60], Alan Lerner[61], Paula Ogrocki[61], Leon Hudson[61], Evan Fletcher[10], Owen Carmichael[10], John Olichney[10], Charles DeCarli[10], Smita Kittur[62], Michael Borrie[63], T-Y Lee[63], Rob Bartha[63], Sterling Johnson[64], Sanjay Asthana[64], Cynthia M. Carlsson[64], Steven G. Potkin[28], Adrian Preda[28], Dana Nguyen[28],

Pierre Tariot[30], Stephanie Reeder[30], Vernice Bates[65], Horacio Capote[65], Michelle Rainka[65], Douglas W. Scharre[66], Maria Kataki[66], Anahita Adeli[66], Earl A. Zimmerman[67], Dzintra Celmins[67], Alice D. Brown[67], Godfrey D. Pearlson[68], Karen Blank[68], Karen Anderson[68], Robert B. Santulli[69], Tamar J. Kitzmiller[69], Eben S. Schwartz[69], Kaycee M. Sink[70], Jeff D. Williamson[70], Pradeep Garg[70], Franklin Watkins[70], Brian R. Ott[71], Henry Querfurth[71], Geoffrey Tremont[71], Stephen Salloway[72], Paul Malloy[72], Stephen Correia[72], Howard J. Rosen[4], Bruce L. Miller[4], Jacobo Mintzer[73], Kenneth Spicer[73], David Bachman[73], Elizabether Finger[55], Stephen Pasternak[55], Irina Rachinsky[55], Dick Drost[55], Nunzio Pomara[74], Raymundo Hernando[74], Antero Sarrael[74], Susan K. Schultz[75], Laura L. Boles Ponto[75], Hyungsub Shim[75], Karen Elizabeth Smith[75], Norman Relkin[17], Gloria Chaing[17], Lisa Raudin[14,17], Amanda Smith[76], Kristin Fargher[76], Balebail Ashok Raj[76], Thomas Neylan[4], Jordan Grafman[21], Melissa Davis[5], Rosemary Morrison[5], Jacqueline Hayes[4], Shannon Finley[4], Karl Friedl[77], Debra Fleischman[39], Konstantinos Arfanakis[39], Olga James[42], Dino Massoglia[73], J. Jay Fruehling[64], Sandra Harding[64], Elaine R. Peskind[25], Eric C. Petrie[66], Gail Li[66], Jerome A. Yesavage[78], Joy L. Taylor[78] & Ansgar J. Furst[78]

[4] UC San Francisco, California 94143, USA. [5] UC San Diego, California 92093, USA. [6] Mayo Clinic, Rochester, New York 14603, USA. [7] UC Berkeley, California 94720, USA. [8] UPenn, Philadelphia, Pennsylvania 19104, USA. [9] USC, Los Angeles, California 90089, USA. [10] UC Davis, California 95616, USA. [11] Brigham and Women's Hospital/Harvard Medical School, Boston, Massachusetts 02115, USA. [12] Indiana University, Bloomington, Indiana 47405, USA. [13] Washington University St Louis, Missouri 63130, USA. [14] Prevent Alzheimer's Disease 2020, Rockville, Maryland 20850, USA. [15] Siemens, Munich 80333, Germany. [16] University of Pittsburgh, Pennsylvania 15260, USA. [17] Cornell University, Weill Cornell Medical College, New York 10065, USA. [18] Albert Einstein College of Medicine of Yeshiva University, Bronx, New York 10461, USA. [19] AD Drug Discovery Foundation, New York City, New York 10019, USA. [20] Acumen Pharmaceuticals, Livermore, California 94551, USA. [21] Northwestern University, Evanston and Chicago, Illinois 60208, USA. [22] National Institute of Mental Health, Rockville, Maryland 20852, USA. [23] Brown University, Providence, Rhode Island 02912, USA. [24] Eli Lilly, Indianapolis, Indiana 46225, USA. [25] University of Washington, Seattle, Washington 98195, USA. [26] University of London, London WC1E 7HU, UK. [27] UCLA, Los Angeles, California 90095, USA. [28] University of Michigan, Ann Arbor, Michigan 48109, USA. [29] University of Utah, Salt Lake City, Utah 84112, USA. [30] Banner Alzheimer's Institute, Phoenix, Arizona 85006, USA. [31] UC Irvine, Irvine, California 92697, USA. [32] National Institute on Aging, Bethesda, Maryland 20892, USA. [33] Johns Hopkins University, Baltimore, Maryland 21218, USA. [34] Richard Frank Consulting, Washington, DC 20001, USA. [35] Oregon Health and Science University, Portland, Oregon 97239, USA. [36] Baylor College of Medicine, Houston, Texas 77030, USA. [37] University of Alabama, Birmingham, Alabama 35233, USA. [38] Mount Sinai School of Medicine, New York City, New York 10029, USA. [39] Rush University Medical Center, Chicago, Illinois 60612, USA. [40] Wien Center, Miami, Florida 33140, USA. [41] New York University, New York City, New York 10003, USA. [42] Duke University Medical Center, Durham, North Carolina 27710, USA. [43] University of Kentucky, Lexington, Kentucky 40506, USA. [44] University of Rochester Medical Center, Rochester, New York 14642, USA. [45] University of Texas Southwestern Medical School, Dallas, Texas 75390, USA. [46] Emory University, Atlanta, Georgia 30322, USA. [47] University of Kansas, Medical Center, Kansas City, Kansas 66103, USA. [48] Mayo Clinic, Jacksonville, Florida 32224, USA. [49] Yale University School of Medicine, New Haven, Connecticut 06510, USA. [50] McGill University/Montreal-Jewish General Hospital, Montreal, Quebec H3T 1E2, Canada. [51] Sunnybrook Health Sciences, Toronto, Ontario M4N 3M5, Canada. [52] University of British Columbia Clinic for AD & Related Disorders, Vancouver, British Columbia V6T 1Z3, Canada. [53] Cognitive Neurology-St Joseph's Health Care, London, Ontario N6A 4V2, Canada. [54] Cleveland Clinic Lou Ruvo Center for Brain Health, Las Vegas, Nevada 89106, USA. [55] St Joseph's Health Care, London, Ontario N6A 4V2, Canada. [56] Premiere Research Institute, Palm Beach Neurology, Miami, Florida 33407, USA. [57] Georgetown University Medical Center, Washington, DC 20007, USA. [58] Banner Sun Health Research Institute, Sun City, Arizona 85351, USA. [59] Boston University, Boston, Massachusetts 02215, USA. [60] Howard University, Washington, DC 20059, USA. [61] Case Western Reserve University, Cleveland, Ohio 20002, USA. [62] Neurological Care of CNY, Liverpool, New York 13088, USA. [63] Parkwood Hospital, London, Ontario N6C 0A7, Canada. [64] University of Wisconsin, Madison, Wisconsin 53706, USA. [65] Dent Neurologic Institute, Amherst, New York 14226, USA. [66] Ohio State University, Columbus, Ohio 43210, USA. [67] Albany Medical College, Albany, New York 12208, USA. [68] Hartford Hospital, Olin Neuropsychiatry Research Center, Hartford, Connecticut 06114, USA. [69] Dartmouth-Hitchcock Medical Center, Lebanon, New Hampshire 03766, USA. [70] Wake Forest University Health Sciences, Winston-Salem, North Carolina 27157, USA. [71] Rhode Island Hospital, Providence, Rhode Island 02903, USA. [72] Butler Hospital, Providence, Rhode Island 02906, USA. [73] Medical University South Carolina, Charleston, South Carolina 29425, USA. [74] Nathan Kline Institute, Orangeburg, New York 10962, USA. [75] University of Iowa College of Medicine, Iowa City, Iowa 52242, USA. [76] University of South Florida: USF Health Byrd Alzheimer's Institute, Tampa, Florida 33613, USA. [77] Department of Defense, Arlington, Virginia 22350, USA. [78] Stanford University, Stanford, California 94305, USA

