## [Peer Review File · Nature Communications]

Reviewers' comments:

Reviewer #1 (Remarks to the Author):

Review of Submission: Transferred to Nature Communications.

This paper was previously reviewed for [redacted]. Substantial and very helpful changes have been made in response to reviewers' concerns.

Brief summary of the paper:

This paper uses sequential structural MRI data from ADNI, stratified by diagnostic category, to examine the hypothesis that atrophy in grey matter (GM) in basal forebrain (BF) precedes and predicts atrophy in entorhinal cortex (EC). Analysis was restricted to participants having data at 3 successive annual visits, denoted T1, T2, and T3. The sample was partitioned into healthy controls (HC); MCI who did not progress to AD (prefer this term to conversion), denoted MNC; MCI who progressed (MC); and those who had AD. The authors have laid out a careful set of implications of this hypothesis, and report a series of analyses, primarily relatively straightforward linear regression models, testing each potential implication. They contrast their results with those predicted by a model in which both regions atrophy in parallel.

The analyses have generally been quite careful. The authors have addressed virtually all of my concerns in my previous review for [redacted]. Minor details missing from the supplemental material have been added (confirming that only ADNI1 participants were included, and how the time points were chosen for participants other than MC group.) The new analyses that eliminate the need for floor effect were helpful. Also, the explanation of the curious findings in old figure 5A makes sense, and dissociation into amyloid and non-amyloid groups does seem to help.

Overall, I think the revision is very responsive, and the paper contains solid analyses pointing to an interesting and important question. Reviewer 3's very helpful comment on the heterogeneity in the early ADNI groups really helped to clarify some of my observations, also, and the re-analysis sorts this out nicely.

A couple of further, small suggestions:

They state that measures of GM volume are "reliable" - this seems like an odd use of language for comparing time 1 to time 2. In fact, we do see changes, and there hypothesis is precisely that we would see variation between people in amount of change, representing

differences in stage, thus the correlation between T2 and T1 would be reduced to the extent that you don't have a perfect linear relationship predicting T2 from T1. I would just drop this altogether.

Minor:

Line 218 should be different stages, not difference stages.

Reviewer #3 (Remarks to the Author):

The authors have made a strong revision and have responded well to reviews. I am still unclear about the subject groups, and don't find this explained satisfactorily in the methods. If the authors are testing hypotheses related to AD, then the patients with a clinical diagnosis of AD who have normal CSF amyloid levels should be excluded, since they may have a different underlying pathology (e.g., frontotemporal dementia, vascular dementia, hippocampal sclerosis, etc). The same can be said for MCI-if amyloid levels are not below 192, they should be excluded since they likely do not have underlying AD pathology. Thus, the groups of interest in this study-unless I'm misunderstanding something-are amyloid negative (high CSF amyloid) cognitively normal older adults (who are the ultimate control group and would not be expected to show much progressive atrophy of these structures), amyloid positive controls (who are considered to be at the earliest stage of preclinical AD), amyloid positive MCI (non-progressors during this time interval), amyloid positive MCI progressors, and amyloid positive AD dementia patients.

I find the first section of the results of questionable value. Why is it useful to look at the measures and their change in the pooled sample as a whole? I do not find Fig 1A to be interpretable and would suggest that it be changed to show the amyloid positive clinical diagnostic subgroups. When it comes to reliability of measurements, I would start by looking at the cognitively normal amyloid negative subjects for a true estimate (Fig 1B). It would be helpful to report ICC or similar typical reliability metric without covariate adjustment, since the same measure in the same individual should not be confounded by covariates.

I think the section on parallel vs predictive spread should come after the basic reporting of the subgroup results as mentioned above. I do not see the value of doing this analysis with the entire sample, which the authors should clarify if they do.

In the section "Linking GM degeneration to CSF biomarkers of AD pathology," the authors report on comparisons between amyloid positive vs amyloid negative groups. This is not particularly useful in my opinion because the hypotheses being tested are focused on AD. People with clinical diagnoses of MCI or AD who are amyloid negative probably do not in fact have the correct diagnosis. I think this entire section should be removed or justified. The following statement: "Across individuals, we therefore expected that decreased CSF levels of amyloid would predict increased magnitudes of GM degeneration." Is correct if the hypothesis be tested is that people with AD have GM degeneration and people without AD

do not, or do not have as much, but that is not the topic of this study.

In the section "Biomarker staging of AD pathology," the authors perform what I would consider to be the most appropriate analysis to test their major hypotheses. I believe this should be the primary section of the results. However, I do not agree with the median split analysis based on CSF a-beta. There are well-established cutoffs used by many investigators to dichotomize people into those likely with amyloid (<192) and those likely without (>192). Those cognitively normal adults likely without amyloid would make an excellent control group whose volumetrics would likely not change appreciably in 3 years. Those CN adults with amyloid would be the earliest detectable stage of Preclinical AD. With regard to MCI and AD, there are certainly members of both groups who are not below 192, indicating the possibility of an incorrect diagnosis (ie MCI or dementia due to a non AD pathophysiology). In ADNI 1, for example, there are a substantial number of MCI patients who have CSF abeta greater than 192, and at least some of them would likely be categorized as progressors in the present analysis. And in an early batch, of the 102 AD dementia patients with CSF, 8 of them were above the cutoff, suggesting the possibility that their dementia that may have appeared to be AD clinically may not actually be AD at a pathophysiological level.

In several of the figures the white bars are difficult to see because the lines around them are too fine (small). Please enlarge those lines or provide shading so those bars and their respective error bars are more visible.

In Fig 1, rather than showing the entire sample, it would be helpful to show the major subgroups separately. That is, amyloid negative (high CSF amyloid) cognitively normal older adults (who would not be expected to show much progressive atrophy of these structures). Amyloid positive controls, amyloid positive MCI (progressors vs non-progressors), and amyloid positive AD.

In Suppl Fig 2, it would be helpful to separate the diagnostic groups into amyloid negative vs. amyloid positive (another figure showing this would be helpful, since this figure based on clinical diagnoses is helpful in itself). For each of these two figures, it would be helpful to provide Cohen's d effect sizes comparing each of the diagnostic groups to the amyloid negative cognitively normal controls.

Finally, in the abstract, the authors state that they examined the spectrum including "advanced AD." In fact, the AD patients in the dementia category have mild dementia, not moderate or severe dementia. This should be rephrased. ADNI was meant to capture the earliest phases of preclinical, prodromal, and mild AD dementia.

Reviewer #4 (Remarks to the Author):

This paper tests the hypothesis that neurodegeneration in AD appears earlier in the basal

forebrain than in the entorhinal cortex, the latter being broadly considered one of the earliest sites for neurodegeneration in typical AD. The authors use a careful series of statistical analyses on data derived from the ADNI data set to test the hypothesis, as well as various associated propositions. In particular, they use mediation analysis and conditional process analysis to provide strong evidence that it is the spread of degeneration from the basal forebrain to the entorhinal cortex that leads to memory impairment.

Overall, this is a very nice piece of work. The experiments and arguments are very well thought out and executed. The presentation and illustrations are very good. I see it has been reviewed already, although this is the first time I've seen it; the response to the previous reviews is thoughtful and appropriate. I recommend publication and have only a few very minor additional suggestions:

1. At least the initial analysis (figure 2) to support the hypothesis of earlier degeneration in Ch4 than EC appears to rely on the specific shape of the degeneration curves (the sigmoid shapes in figure 2 A and B). Specifically, it assumes acceleration of degeneration/pathology in the early stages. If the volume loss were linear, we would not expect any significant effect even in the EC -> Ch4 model. Later results I believe do confirm the authors' hypothesis without this assumption, so I don't think this changes any of the key conclusions, but the paper should make the assumption clear and discuss its potential consequences.

2. Also in figure 2 and later figures, a clear definition of "Adjusted X" would be useful - I worked out what it is eventually, but an explicit definition would have helped.

3. Page 9. "... to generate a model for the temporal ordering of AD biomarkers..." Another useful reference here is Young et al Brain 2014 "A data driven model of biomarker changes in sporadic AD", which uses less assumptions confirming the point the authors make here even more strongly.

4. I was hoping the Discussion might discuss potential implications of early involvement of Ch4. For example, does this give any clues as to what symptoms of AD might be observable even earlier than the memory impairment that is mediated by EC degeneration?

Reviewer #1 (Remarks to the Author):

Review of Submission: Transferred to Nature Communications.

This paper was previously reviewed for [redacted]. Substantial and very helpful changes have been made in response to reviewers' concerns.

Brief summary of the paper:

This paper uses sequential structural MRI data from ADNI, stratified by diagnostic category, to examine the hypothesis that atrophy in grey matter (GM) in basal forebrain (BF) precedes and predicts atrophy in entorhinal cortex (EC). Analysis was restricted to participants having data at 3 successive annual visits, denoted T1, T2, and T3. The sample was partitioned into healthy controls (HC); MCI who did not progress to AD (prefer this term to conversion), denoted MNC; MCI who progressed (MC); and those who had AD. The authors have laid out a careful set of implications of this hypothesis, and report a series of analyses, primarily relatively straightforward linear regression models, testing each potential implication. They contrast their results with those predicted by a model in which both regions atrophy in parallel.

We greatly appreciate reviewer 1's positive response. The feedback in the prior and current revision was exceptionally useful and has, in our mind, greatly strengthened the manuscript. We agree that the MCI group labels implying that AD is a 'conversion' process is less accurate than 'progression.' We have thus re-named our MCI groups: MCI non-progressors (MCI-NP) and MCI who progressed to AD (MCI-AD).

The analyses have generally been quite careful. The authors have addressed virtually all of my concerns in my previous review for [redacted]. Minor details missing from the supplemental material have been added (confirming that only ADNI1 participants were included, and how the time points were chosen for participants other than MC group.) The new analyses that eliminate the need for floor effect were helpful. Also, the explanation of the curious findings in old figure 5A makes sense, and dissociation into amyloid and non-amyloid groups does seem to help.

Overall, I think the revision is very responsive, and the paper contains solid analyses pointing to an interesting and important question. Reviewer 3's very helpful comment on the heterogeneity in the early ADNI groups really helped to clarify some of my observations, also, and the re-analysis sorts this out nicely.

A couple of further, small suggestions:

They state that measures of GM volume are "reliable" - this seems like an odd use of language for comparing time 1 to time 2. In fact, we do see changes, and there hypothesis is precisely that we would see variation between people in amount of change, representing differences in stage, thus the correlation between T2 and T1 would be reduced to the extent that you don't have a perfect linear relationship predicting T2 from T1. I would just drop this altogether.

The measures of test-retest reliability in Figure 1b were originally intended to demonstrate that there was low test-retest variability in the ADNI scans over time points, e.g. due to scan artefacts in the image volumes or suboptimal nonlinear warping during pre-processing. However, the reviewer is correct to point out that these correlations should not be perfect in any event, given that volumes are changing (decreasing) within subjects between time points. Indeed, we observed that all of the inter-scan correlations are shifted off-diagonal in the same direction, indicating smaller volumes at the second time point. We therefore agree with reviewer 3 that this analysis is slightly misleading. Moreover, it is largely unnecessary, given that the ADNI MRI dataset has already been subjected to rigorous data quality checks and filters. We have therefore removed these analyses from the text and from Figure 1. See section in lines 85—103.

Minor:

Line 218 should be different stages, not difference stages.

We have corrected this typo accordingly (line 58).

Reviewer #3 (Remarks to the Author):

The authors have made a strong revision and have responded well to reviews. I am still unclear about the subject groups, and don't find this explained satisfactorily in the methods. If the authors are testing hypotheses related to AD, then the patients with a clinical diagnosis of AD who have normal CSF amyloid levels should be excluded, since they may have a different underlying pathology (e.g., frontotemporal dementia, vascular dementia, hippocampal sclerosis, etc). The same can be said for MCI-if amyloid levels are not below 192, they should be excluded since they likely do not have underlying AD pathology. Thus, the groups of interest in this study-unless I'm misunderstanding something-are amyloid negative (high CSF amyloid) cognitively normal older adults (who are the ultimate control group and would not be expected to show much progressive atrophy of these structures), amyloid positive controls (who are considered to be at the earliest stage of preclinical AD), amyloid positive MCI (non-progressors during this time interval), amyloid positive MCI progressors, and amyloid positive AD dementia patients.

This paper has improved substantially with reviewer 3's meticulous feedback in the prior version, in particular the suggestion for the ABETA analysis, which we believe plays a pivotal role in establishing the evidence for Ch4→EC pathological staging. We agree with the reviewer that there is ambiguity in the underlying pathology of clinical patients with high ABETA. We address this important issue below, where Reviewer 3 further elaborates it.

I find the first section of the results of questionable value. Why is it useful to look at the measures and their change in the pooled sample as a whole? I do not find Fig 1A to be interpretable and would suggest that it be changed to show the amyloid positive clinical diagnostic subgroups. When it comes to reliability of measurements, I would start by looking at the cognitively normal amyloid negative subjects for a true estimate (Fig 1B). It would be helpful to report ICC or similar typical reliability metric without covariate adjustment, since the same measure in the same individual should not be confounded by covariates.

In general, the first part of the paper utilizes the entire sample of 434 who have three MRI time-points of data available through ADNI (as specified in lines 63, 108). The second part of the paper presents a subset of this sample (N=216), who have both the three MRI time-points and CSF ABETA data available through ADNI (as specified in lines 185, 272). We have organized the paper in this way in order to utilize the full dataset, while still capitalizing on the CSF biomarker for more refined analysis. We feel that this juxtaposition of the samples underscores the importance of using a multi-modal analytical strategy to leverage biomarker sensitivity.

In this revision, we have made every attempt to incorporate reviewer 3's suggestions within this organizational framework.

Based on reviewer 3's feedback, we have changed Figure 1 to show the effect of clinical diagnosis (HC, MCI-NP, MCI-AD, AD) on both GM degeneration [T1 – T3], and baseline volume [T1] in the full sample (N=424). Based on reviewer 1's feedback, we have removed the non-essential reliability measures.

I think the section on parallel vs predictive spread should come after the basic reporting of the subgroup results as mentioned above. I do not see the value of doing this analysis with the entire sample, which the authors should clarify if they do.

The first section on parallel versus predictive spread (e.g. Figure 3) based on the entire sample (as specified on line 108) serves to juxtapose against the last section, where we explore predictive spread on the subsample with both MRI and CSF ABETA (as specified on line 272; e.g. Figure 4). In this latter section, we show that CSF ABETA moderates the Ch4→EC predictive spread pattern. Again, we feel that the full sample and ABETA subsample analyses of predictive pathological spread complement one another.

In the section "Linking GM degeneration to CSF biomarkers of AD pathology," the authors report on comparisons between amyloid positive vs amyloid negative groups. This is not particularly useful in my opinion because the hypotheses being tested are focused on AD. People with clinical diagnoses of MCI or AD who are amyloid negative probably do not in fact have the correct diagnosis. I think this entire section should be removed or justified. The following statement: "Across individuals, we therefore expected that decreased CSF levels of amyloid would predict increased magnitudes of GM degeneration." Is correct if the hypothesis be tested is that people with AD have GM degeneration and people without AD do not, or do not

have as much, but that is not the topic of this study.

In retrospect, we agree with reviewer 3 that this section does not add conceptually to the central theses of the paper. We have thus removed the whole group CSF/GM regression analyses from this section, as well as its separate heading. What remains is now used as the opening paragraph for the ABETA clinical staging section, where we define our samples (see lines 165—191).

We also agree that inclusion of high ABETA clinical patients is problematic and may be misdiagnosed. We now exclude from all subsequent analyses the MCI-NC, MCI-AD, and AD individuals with normal CSF ABETA levels (as specified on lines 181—186)).

In the section "Biomarker staging of AD pathology," the authors perform what I would consider to be the most appropriate analysis to test their major hypotheses. I believe this should be the primary section of the results. However, I do not agree with the median split analysis based on CSF a-beta. There are well-established cutoffs used by many investigators to dichotomize people into those likely with amyloid (<192) and those likely without (>192). Those cognitively normal adults likely without amyloid would make an excellent control group whose volumetrics would likely not change appreciably in 3 years. Those CN adults with amyloid would be the earliest detectable stage of Preclinical AD.

We have analyzed the volumetric data both ways, using a median split or the 192 cutoff, and the results are the same in either case. See below:

Based upon the reviewer's feedback however, we agree that it is more compelling and more reproducible to use the established cut-points from the Shaw et al (2009) paper. All analyses in the paper utilizing the CSF ABETA measures now adhere to the 192 cutpoint.

With regard to MCI and AD, there are certainly members of both groups who are not below 192, indicating the possibility of an incorrect diagnosis (ie MCI or dementia due to a non AD pathophysiology). In ADNI 1, for example, there are a substantial number of MCI patients who have CSF abeta greater than 192, and at least some of them would likely be categorized as

progressors in the present analysis. And in an early batch, of the 102 AD dementia patients with CSF, 8 of them were above the cutoff, suggesting the possibility that their dementia that may have appeared to be AD clinically may not actually be AD at a pathophysiological level.

This is an excellent point. As mentioned above, we have removed the following individuals from all analyses involving CSF ABETA (as specified on lines 181—186):

MNC ABETA > 192 (N=19)

MC ABETA > 192 (N=4)

AD ABETA > 192 (N=5)

TOTAL = 28

Original ABETA total sample = 244

New ABETA total sample = 216

We appreciate this and Reviewer 3's comments to refine our results specifically to AD pathogenesis.

In several of the figures the white bars are difficult to see because the lines around them are too fine (small). Please enlarge those lines or provide shading so those bars and their respective error bars are more visible.

We have removed all figures with white bars. Line weights were increased in remaining figures to darken contours.

In Fig 1, rather than showing the entire sample, it would be helpful to show the major subgroups separately. That is, amyloid negative (high CSF amyloid) cognitively normal older adults (who would not be expected to show much progressive atrophy of these structures). Amyloid positive controls, amyloid positive MCI (progressors vs non-progressors), and amyloid positive AD.

This is now exactly what is portrayed in Figure 3b and in the text. We believe this juxtaposes well with Figure 1 now, which we have also modified to show GM degeneration as a function solely of clinical diagnosis in the whole sample (N=434).

In Suppl Fig 2, it would be helpful to separate the diagnostic groups into amyloid negative vs. amyloid positive (another figure showing this would be helpful, since this figure based on clinical diagnoses is helpful in itself). For each of these two figures, it would be helpful to provide Cohen's d effect sizes comparing each of the diagnostic groups to the amyloid negative cognitively normal controls.

As mentioned above, we now show GM degeneration as a function of ROI and clinical diagnosis in the full sample (N=434) in Figure 1a. We then show the extension of this analysis in the multimodal subsample (N=216), separating clinical groups according to the ABETA cutpoint, in Figure 3. We now report Cohen's d effect sizes for the critical independent samples t-tests comparing the early stage diagnostic groups (HC-AB+, MCI-NP-AB+). Our effect sizes are moderate (~0.50) according to Cohen's (1988) conventions. See lines 199—236.

Finally, in the abstract, the authors state that they examined the spectrum including "advanced AD." In fact, the AD patients in the dementia category have mild dementia, not moderate or severe dementia. This should be rephrased. ADNI was meant to capture the earliest phases of preclinical, prodromal, and mild AD dementia.

We thank the reviewer for pointing this out. We have rephrased the abstract to simply 'AD.' See line 3.

Reviewer #4 (Remarks to the Author):

This paper tests the hypothesis that neurodegeneration in AD appears earlier in the basal forebrain than in the entorhinal cortex, the latter being broadly considered one of the earliest sites for neurodegeneration in typical AD. The authors use a careful series of statistical analyses on data derived from the ADNI data set to test the hypothesis, as well as various associated propositions. In particular, they use mediation analysis and conditional process analysis to provide strong evidence that it is the spread of degeneration from the basal forebrain to the entorhinal cortex that leads to memory impairment.

Overall, this is a very nice piece of work. The experiments and arguments are very well thought out and executed. The presentation and illustrations are very good. I see it has been reviewed already, although this is the first time I've seen it; the response to the previous reviews is thoughtful and appropriate. I recommend publication and have only a few very minor additional suggestions:

1. At least the initial analysis (figure 2) to support the hypothesis of earlier degeneration in Ch4 than EC appears to rely on the specific shape of the degeneration curves (the sigmoid shapes in figure 2 A and B). Specifically, it assumes acceleration of degeneration/pathology in the early stages. If the volume loss were linear, we would not expect any significant effect even in the EC -> Ch4 model. Later results I believe do confirm the authors' hypothesis without this assumption, so I don't think this changes any of the key conclusions, but the paper should make the assumption clear and discuss its potential consequences.

We appreciate the reviewer's receptiveness to the paper. The sigmoid functions of degeneration is derived from a theory paper by Jack et al. (2010) and several empirical papers, all of which we now explain and cite (see lines 127—131).

2. Also in figure 2 and later figures, a clear definition of "Adjusted X" would be useful - I worked out what it is eventually, but an explicit definition would have helped.

We have removed the 'Adjusted' from the axis labels as it is indeed distracting. We state in each figure caption that the regression coefficients are adjusted, and for what they are adjusted.

3. Page 9. "... to generate a model for the temporal ordering of AD biomarkers..." Another useful reference here is Young et al Brain 2014 "A data driven model of biomarker changes in sporadic AD", which uses less assumptions confirming the point the authors make here even more strongly.

We are sorry to have missed this paper in our initial literature review. We have added this citation (see line 173)

4. I was hoping the Discussion might discuss potential implications of early involvement of Ch4. For example, does this give any clues as to what symptoms of AD might be observable even earlier than the memory impairment that is mediated by EC degeneration?

We have some ideas about this, which were somewhat obliquely mentioned in the introduction. We have added a more thorough treatment to the potential perceptual and behavioural manifestations of cholinergic Ch4 pathology in the discussion (see lines 325—336).

REVIEWERS' COMMENTS:

Reviewer #3 (Remarks to the Author):

The authors have responded adequately to my concerns, with the exception of two small additional minor points.

1. Is the F value for Ch4 in this sentence incorrectly stated?

Magnitudes of GM degeneration in both Ch4 and EC were significantly impacted by diagnosis (Ch4: $F_{3,425}=5.33$, $p=0.001$; EC: $F_{3,425}=53.39$, $p<0.001$)

2. The authors do not clearly state as a limitation in the discussion the relatively small number of participants in the overall study with amyloid biomarkers; the study would clearly be stronger, but not as possible to model statistically, if they only include the participants with biomarkers supportive of AD pathology. I understand that the full sample is necessary to carry out robust statistical analyses, but it is a limitation to not have biomarker data on everyone. This should be stated.

REVIEWERS' COMMENTS:

Reviewer #3 (Remarks to the Author):

The authors have responded adequately to my concerns, with the exception of two small additional minor points.

1. Is the F value for Ch4 in this sentence incorrectly stated?

Magnitudes of GM degeneration in both Ch4 and EC were significantly impacted by diagnosis (Ch4: $F_{3,425}=5.33$, $p=0.001$; EC: $F_{3,425}=53.39$, $p<0.001$)

We have checked over the statistics for these models and confirmed that they are accurate. The attenuation of the F-statistic for the Ch4 model is due to the fact the group differences are smaller at more advanced stages of disease (e.g. MCI-AD versus AD). However, we did observe that we misreported the primary Group X ROI interaction effect for the [T1 – T3] longitudinal model. The prior statistics were derived from a model with only Ch4 and EC included (hence the interaction degrees of freedom were also incorrect at 3,425). We have corrected this so that the model includes all three ROIs: Ch4, EC, and PSC (line 106). The results are virtually identical. We have checked over our statistics throughout to ensure they are correct.

2. The authors do not clearly state as a limitation in the discussion the relatively small number of participants in the overall study with amyloid biomarkers; the study would clearly be stronger, but not as possible to model statistically, if they only include the participants with biomarkers supportive of AD pathology. I understand that the full sample is necessary to carry out robust statistical analyses, but it is a limitation to not have biomarker data on everyone. This should be stated.

We acknowledge that the healthy older adult sub-samples are relatively small. This was the maximum sample size we could obtain through the ADNI that have multimodal biomarkers traversing CSF A β , three time points of structural MRI, and three time points of neuropsych. All three of these measures are necessary to triangulate the patterns we observed in this study. As requested by the reviewer, we have included the following statement in the discussion (lines 405-410).

“One limitation of this study is the relatively small number of HCA β + individuals available through ADNI whose measures of CSF A β can be integrated with longitudinal structural MRI and neuropsychological data (n=28). Our findings indicate that future large-scale research initiatives on AD would benefit from a multimodal biomarker strategy including, at a minimum, CSF A β and longitudinal structural MRI, focused on cognitively healthy adults.”